# Phosphate Capture Enhancement Using Designed Iron Oxide-Based Nanostructures

**DOI:** 10.3390/nano13030587

**Published:** 2023-02-01

**Authors:** Paula Duenas Ramirez, Chaedong Lee, Rebecca Fedderwitz, Antonia R. Clavijo, Débora P. P. Barbosa, Maxime Julliot, Joana Vaz-Ramos, Dominique Begin, Stéphane Le Calvé, Ariane Zaloszyc, Philippe Choquet, Maria A. G. Soler, Damien Mertz, Peter Kofinas, Yuanzhe Piao, Sylvie Begin-Colin

**Affiliations:** 1Institut de Physique et Chimie des Matériaux de Strasbourg, UMR 7504, University of Strasbourg, CNRS, 67034 Strasbourg, France; 2Graduate School of Convergence Science and Technology, Seoul National University, 145 Gwanggyo-ro, Yeongtong-gu, Suwon-Si 16229, Gyeonggi-do, Republic of Korea; 3Department of Chemical and Biomolecular Engineering, University of Maryland, 4418 Stadium Dr., College Park, MD 20740, USA; 4Institute of Physics, University of Brasilia, Brasilia 70910900, Brazil; 5Institut de Chimie et Procédés pour l’Energie, l’Environnement et la Santé (ICPEES), UMR-7515 CNRS-Université de Strasbourg, 25 rue Becquerel, 67087 Strasbourg, France; 6Laboratoire des Sciences de l’Ingénieur, de l’Informatique et de l’Imagerie (ICube)—CNRS/University of Strasbourg, UMR 7357 Preclinical Imaging Lab, Imaging Dpt, Hôpitaux Universitaires de Strasbourg, 67098 Strasbourg, France; 7Advanced Institutes of Convergence Technology, 145 Gwanggyo-ro, Yeongtong-gu, Suwon-si 16229, Gyeonggi-do, Republic of Korea

**Keywords:** iron oxide nanoclusters, iron precursor effect, aluminium, zinc and cobalt doping, phosphate adsorption studies

## Abstract

Phosphates in high concentrations are harmful pollutants for the environment, and new and cheap solutions are currently needed for phosphate removal from polluted liquid media. Iron oxide nanoparticles show a promising capacity for removing phosphates from polluted media and can be easily separated from polluted media under an external magnetic field. However, they have to display a high surface area allowing high removal pollutant capacity while preserving their magnetic properties. In that context, the reproducible synthesis of magnetic iron oxide raspberry-shaped nanostructures (RSNs) by a modified polyol solvothermal method has been optimized, and the conditions to dope the latter with cobalt, zinc, and aluminum to improve the phosphate adsorption have been determined. These RSNs consist of oriented aggregates of iron oxide nanocrystals, providing a very high saturation magnetization and a superparamagnetic behavior that favor colloidal stability. Finally, the adsorption of phosphates as a function of pH, time, and phosphate concentration has been studied. The undoped and especially aluminum-doped RSNs were demonstrated to be very effective phosphate adsorbents, and they can be extracted from the media by applying a magnet.

## 1. Introduction

Phosphates released into the environment come from agricultural (fertilizers) and industrial sources, from human excreta, and from phosphate-based detergents or washing powder. In Western Europe, point-source phosphate pollution is estimated at 3.5 grams per capita per day: 1.2 grams from human excreta, and the rest mainly from detergents. Moreover, 0.5 to 2.5% of phosphorus used in fertilizers is washed away from cultivated soils by rain and drainage water [1]. Phosphates are the main cause of eutrophication and dystrophication in France and worldwide. Although they are not toxic in themselves for animal and plant life, they are harmful to the environment when present in high concentrations: they become real fertilizers for aquatic environments, which they help enrich excessively in organic matter. For instance, the algae growth induced by phosphorus makes waters less attractive for swimming and other aquatic recreation and degrades the conditions that fish, bugs, wildlife, and desired plants need to grow. In addition to these pollution problems, phosphorus remains a scarce resource, and world reserves of phosphates are limited [2]. A shortage of phosphate fertilizers would have significant consequences for world food production. It is therefore urgent to find solutions for phosphorus recovery. 

Iron oxide nanoparticles have been widely developed to remove phosphates from polluted media [3,4,5,6] and appear to be very suitable compounds to enhance the phosphate removal from polluted media due to the high affinity of phosphate for iron. In addition, iron oxide nanoparticles may be recycled. Indeed, an earlier study using iron oxide nanoparticles of 40 nm demonstrated that it was possible to recycle iron oxide nanoparticles by adjusting pH, and the performance was maintained after several cycles [7]. The effective recycling implies that the nanoparticles can be easily separated from polluted media by an external magnetic field. For such magnetic extraction, the nanoparticles have to display a high saturation magnetization, but at the nanoscale, the saturation magnetization of iron oxide nanoparticles is strongly decreased due to spin canting and defect effects [8,9,10]. At the same time, the nanoparticles have to display a high surface area that would allow high removal pollutant capacity. Therefore, a balance between high saturation magnetization and high surface specific area has to be found. Iron oxide raspberry-shaped nanostructures (RSNs), which consist of oriented aggregates/nanoclusters of iron oxide nanograins, possess very high saturation magnetization close to that of the bulk magnetite phase (85 emu/g), which makes them easy to recover from liquid media with a magnet. In addition, these nanostructures are synthesized by a modified polyol-solvothermal process, which allows for the production of large amounts of powder [11].

**Table 1 nanomaterials-13-00587-t001:** Influence of the substituents on magnetite adsorption capacity towards phosphates in aqueous media.

Substituents	Adsorption Capacity of Magnetite (mg·g^−1^)	Adsorption Capacity of Doped Magnetite (mg·g^−1^)	Increase in the Adsorption Capacity (%)	Ref.
**Aluminum**	1.21	5.96	492	[4]
**Zinc-aluminum**	/	21.8	/	[12]
**Magnesium-aluminum**	/	20.0	/	
**Nickel-aluminum**	/	17.4	/	
**Silica-lanthanum oxide**	11.02	27.8	252	[13]
**Aluminum-doped ferrite**	/	43.9	/	[14]
**Cobalt ferrite and magnetite**	1.16	1.05	/	[15]
**Fe-Cu binary oxide**		35.2	/	[16]
**Fe_3_O_4_@C@ZrO_2_**	5.2	13.99	/	[17]
**Fe-Zr binary oxide**	5	13.65	/	
	3.2	21.3	/	

Furthermore, it was demonstrated that doping of iron oxide is also a promising approach to improve phosphate uptake [4,13,14,15,16,17,18]. Table 1 presents some studies on how the doping of spinel iron oxide nanoparticles may increase the phosphate adsorption capacity. However, the many missing data points in the literature on the adsorption capacity of undoped spinel iron oxide make it difficult to estimate the increase in adsorption capacity. In addition, the homogeneous doping of iron oxide nanostructures synthesized by a polyol-solvothermal approach remains a chlenge. Nevertheless, among doping elements, aluminum appears promising and was thus selected to dope RSNs to enhance phosphate removal [4,12].

In this work, the reproducibility of the synthesis of raspberry-shaped iron oxide nanostructures (RSNs) described by Gerber et al. [11] was first studied, followed by the doping of RSNs by zinc, cobalt, and aluminum. We demonstrated the strong impact of starting iron chloride precursors and of the mixing and reaction times on the nanostructure characteristics (diameter, nanograin size, doping efficiency). Then, the phosphate uptake of RSNs and aluminum-doped RSNs in water was compared. Figure 1 summarizes the different studies on the RSN and the performed experiments on the RSNs and aluminum-doped RSNs (Al-RSNs) for phosphate removal.

## 2. Materials and Methods

### 2.1. Materials

Iron(III) chloride hexahydrate (CAS: 10025-77-1), urea 99.3+% (CAS: 57-13-6), ethylene glycol 99% (CAS: 107-21-1), trisodium citrate dihydrate 99% (CAS: 6132-04-3), ammonium molybdate 99% (CAS 13106-76-8), zinc(II) chloride 98% (CAS 7646 85 7), and cobalt(II) chloride hexahydrate 98% (CAS 7791-13-1) were supplied by Alfa Aesar. Succinic acid (CAS: 110-15-6), ammonium hydroxide solution 25% (CAS: 1336-21-6), and nickel(II) chloride 98% (CAS 7718-54-9) were provided by Sigma-Aldrich, and cetyltrimethylammonium bromide 99% (CTAB) (CAS: 57-09-0) by Roth. Deionized water was used for all experiments. 

### 2.2. Synthesis Conditions of Iron Oxide Based RSNs

**Magnetite Fe_3−x_O_4_ RSN synthesis.** In a typical synthesis, FeCl_3_·6H_2_O (1.626 g), succinic acid (0.24 g), and urea (3.6 g) were completely dissolved in ethylene glycol (EG) (60 mL) by vigorous mechanical mixing (700 rpm) (for 2 hours up to overnight) and sonication (3 times 20 minutes, changing the water of the ultrasound bath) (Figure 2). The solution was carefully sealed in a Teflon-lined stainless steel autoclave (75 mL capacity) and slowly heated at 200 °C at a heating rate of 1.5 °C/min. The temperature was kept at 200 °C for several hours (between 5 and 10.5 h). The autoclave was then cooled down to room temperature outside of the oven for 3 hours. The black sediment was separated from the supernatant by magnetic decantation. In the first trial, it was washed three times with ethanol and three times with deionized water to eliminate organic and inorganic impurities. This washing step has been improved here and will be discussed later.

Systematically, before each synthesis experiment, the glasses and Teflon containers were cleaned with an acidic solution (HCl, 6 M) for at least one hour and then washed with deionized water and ethanol. 

The optimized parameters were the following: reactants dissolution step by adjusting the mixing time (3 h, 8 h or overnight, 24 h), the reaction time (6 or 10.5 h), the autoclave size, the cooling protocol of the autoclave (inside or outside the oven).

**Doped RSNs.** The same protocol as for the iron oxide RSNs was kept (Figure 2). Zinc chloride (ZnCl_2_·6H_2_O) was used as the precursor for zinc, cobalt, chloride hexahydrate (CoCl_2_·6H_2_O) for cobalt, and aluminum chloride hexahydrate (AlCl_3_·6H_2_O) for aluminum. The synthesis of doped nanostructures by applying the same synthesis protocol as for undoped iron oxide RSNs has not been successful. We have therefore investigated this doping step by carrying out the synthesis by substituting cobalt to iron in different 1:X ratios (X = 1, 2, 3, 4, 7, 9—depending on the synthesis). The total concentration of metal cations was kept at 0.1 mol·L^−1^.

The synthesis conditions for each doping element are given below.

***Zinc-doped RSNs.*** Mixing time of 3 h and reaction time of 12 h. A long reaction time is needed to ensure the dispersion of the Zn inside the structure. Indeed, a study by Nguyen et al. [19] reported on the formation, at first, of a Zn shell around the RSNs, with Zn gradually diffusing when increasing the reaction time.

***Cobalt-doped RSNs.*** Mixing time of 3 h or overnight, and a reaction time of 10.5, 21, or 30 h. The two different mixing times evidenced the importance of a good dissolution of CoCl_2_·6H_2_O in the reaction mixture.

***Aluminum-doped RSNs.*** Due to the similarities between Fe^3+^ and Al^3+^ cations (same valence, similar ionic sizes, same hydration), we kept a mixing time of 3 h and a reaction time of 10.5 h. 

***Washings of RSNs.*** After their synthesis, the RSNs were separated from the supernatant by magnetic decantation and dispersed in a mixture of 50/50 ethanol and warm acetone (60 °C). The suspension was ultrasonicated for 5 min, then magnetically decanted to remove the supernatant. Different washing treatments were tested, but traces of reactants (especially EG) always remained at the RSN surface. To remove the most of EG traces, the RSNs were washed 9–10 times.

### 2.3. Phosphate Adsorption Experiments

#### 2.3.1. Preparation of Phosphates Solutions

Orthophosphoric acid was used as a phosphate source. A first solution at 309.9 P-mg/L was freshly prepared by introducing 68.4 µL of orthophosphoric acid (85%) in 100 ml of Milli-Q water. The solution was adjusted using a 2 M NaOH aqueous solution to precisely set pH 7. Other phosphate solutions were prepared by diluting this solution.

#### 2.3.2. Adsorption Experiments

The RSN suspension was added to a solution containing phosphate at a known concentration. After magnetic mixing for different times for the kinetics study and for 24 h for the adsorption study, the RSNs were separated by magnetic decantation and washed with water. Finally, the phosphate remaining in supernatants was analyzed by UV–vis spectrophotometry using the “blue of *molybdene*” *method* described below.

***The blue of molybdenum method.*** This method was already performed by Daou et al. [20] in order to quantify the amount of phosphates in supernatants. This method is effective and allows quantifying very low concentrations of phosphates in water. It consists in generating a phosphate complex, which can be quantified using UV spectroscopy.
7 H_2_PO_4_^−^ + 12 (NH_4_)6Mo_7_O_24_ + 72 H^+^ →7 (NH_4_)H_2_PO_4_(MoO_3_)_12_ + 65 NH_4_^+^ + 36 H_2_O(1)
(NH_4_)H_2_PO_4_(MoO_3_)_12_ + 6 Sn^2+^ → ‘Blue of Molybdenum’ + 6 Sn_4_^+^(2)

*Preparation:* 2.5 g of ammonium molybdate ((NH_4_)_6_Mo_7_O_24_) was dispersed into 17.5 mL of pure water. In parallel, 28 mL of sulfonic acid (H_2_SO_4_, Carlo Erba) was dispersed in 40 mL of ddH_2_O. Both solutions were then mixed and diluted up to 100 mL with ddH_2_O to obtain the first reagent. 

The second reagent was prepared as follows: 0.5 g of tin chloride dihydrate (SnCl_2_·2H_2_O) was dispersed in 50 mL of 85% glycerol. Both reagents were kept in glass containers and wrapped with aluminum paper to avoid premature degradation from light exposure.

Five hundred microliters of the molybdenum reagent and 500 μL of the tin reagent were mixed with the appropriate amount of the phosphate solution and filled up to 10 mL with ddH_2_O.

*Nota Bene:* These experiments have to be carried out under specific conditions: (i) the temperature must be between 20 and 25 °C to ensure the complex formation; (ii) to ensure that no traces of phosphates remain on the glasses, all required glasses are cleaned before each measurement with NaOH 2 M [20]; (iii) the blue of molybdenum complex is not stable with time, requiring a rapid measurement of absorption, typically after 10 min of mixing. Indeed, at the end of the mixing step, the complex is stable for only 15 min.

The reference mixture (obtained without phosphate) also undergoes a form of degradation. Therefore, the reference has to be repeated for each measurement. Regarding the measurement uncertainties, the complexation reaction itself does not always occur at the same reaction rate, leading to different values for a fixed concentration. The precision of the 10 min timing can also generate additional uncertainty.

***Kinetics of phosphate adsorption as a function of time***. For these experiments, a known amount of RSNs was dispersed in a phosphate solution with a given concentration, and the amount of remaining phosphate in the supernatant was analyzed at different times. Experimentally, 20 mg of RSNs was dispersed in 20 mL of a phosphate solution (50 P-mg·L^−1^ at pH 7). The mixture was stirred on a rotating wheel for 30 min up to 24 h.

***Isotherm of adsorption experiments*.** Twenty milligrams of RSNs was dispersed in 20 mL of phosphate solution at different concentrations. The experiments were performed in water at pH 3 and 7. The range of investigated phosphate concentrations was 3.1–154.9 P-mg·L^−1^. The mixtures were stirred on a rotating wheel for 24 h.

In the first approach, we considered the S, L, H, and C isotherm models to understand the interactions between phosphates and iron oxide. Then, different kinetics models were used to characterize more precisely the phosphate adsorption kinetics: the Langmuir, Freundlich, and Redlich–Peterson models.

### 2.4. Characterization Techniques

Thermal gravimetric analysis (TGA) measurements were performed with a TA Instruments DSCQ1000 instrument (New Castle, DE, USA) operated at a scanning rate of 5 °C min^−1^ on heating and cooling.

The NPs were characterized by transmission electron microscopy (TEM) with a JEOL 2100 microscope (Tokyo, Japan) operating at 200 kV (point resolution: 0.18 nm).

The X-ray diffraction (XRD) pattern was collected at room temperature with a Bruker D8 Advance diffractometer equipped with a monochromatic copper radiation source (Kα = 0.154056 nm) and a Lynx-Eye detector in the 25–65°(2θ) range with a scan step of 0.03°. High-purity silicon powder (a = 0.543082 nm) was systematically used as an internal standard. Profile matching refinements were performed through the Fullprof program [21] using Le Bail’s method [22] with the modified Thompson–Cox–Hasting (TCH) pseudo-Voigt profile function.

Standard infrared spectra were recorded between 4000 and 400 cm^−1^ with a Fourier transform infrared (FTIR) spectrometer, (Spectrum 100, Perkin Elmer, Waltham, MA, USA). Samples were gently ground and diluted in non-absorbent KBr matrices.

Magnetic measurements were performed using a superconducting quantum interference device (SQUID) magnetometer (Quantum Design MPMS-XL 5, San Diego, CA, USA). Magnetization curves as a function of a magnetic field (M(H) curves) were measured at 300 K. Magnetization saturation (M_S_) was measured from hysteresis recorded at 300 K and was determined after removing the mass of organic ligands according to TGA experiments.

BET nitrogen (N_2_) adsorption/desorption isotherms. To characterize the surface specific area of the RSNs, adsorption and desorption of nitrogen isotherms were measured on an ASAP 2420 V instrument with around 100 mg of RSNs. The Brunauer–Emmett–Teller (BET) method was used to calculate the surface area. 

## 3. Results

### 3.1. Optimization of the Synthesis of Iron Oxide Nanostructures

#### 3.1.1. Reproducibility of RSN Synthesis

The protocol of Gerber et al. [11] was first reproduced using the same reactants (iron precursor flask: Alfa-Aesar 2 (AA2)) and under the same conditions (Appendix A). The so-synthesized RSNs were characterized with SEM and TEM, FTIR spectroscopy, X-ray diffraction, and magnetic measurements. These characterizations are detailed in the SI and summarized in Appendix A. The structural and magnetic characterizations (Appendix A) confirmed an oriented aggregation of nanograins, but the RSN diameter and nanograin sizes were observed to be different from the study of Gerber et al. [11]. Indeed, using the same reactants and synthesis conditions, the mean diameter of RSNs is smaller (245 vs. 157 nm) when the mean grain size is higher (25 vs. 30 nm). For both RSN samples, the grain size is higher than the crystallite size deduced from the XRD pattern. Gerber et al. [23] attributed such a mismatch between both sizes to the presence of defects or dislocations resulting from the formation mechanism of the oriented aggregates (coalescence of grains, recrystallization processes). The lattice parameter is similar and agrees with the presence of mostly magnetite. Its slightly higher value than the bulk magnetite phase value may be explained by the presence of defects and strains induced by the oriented aggregation. The saturation magnetization (M_S_, _300K_) value is also lower (78 [23] vs. 70 emu·g^−1^ (this study)) and may be explained by the difference in sizes leading to a slightly higher oxidation of the magnetite phase in RSNs. Indeed, Gerber et al. observed similar results in RSNs, with larger nanograins displaying lower interfaces, which normally contribute to preventing nanograins from oxidation.

To conclude, the reactants and experimental conditions were the same, and the only parameter was their “aging” (two years). An impact of the precursor nature on the synthesis of RSNs was at first suspected. Indeed, the iron precursor flask was stored in the laboratory without particular precautions and an effect of the aging of the iron precursor was also suspected. In order to investigate a possible effect of the iron chloride precursor, the RSN synthesis was performed using different FeCl_3_·6H_2_O precursors provided by different companies or stocked at different times in the laboratory.

#### 3.1.2. Effect of the Commercial Nature of Hexahydrate Iron Chloride(III) Precursor

In the following experiments, the sole difference was the brand and batch of hexahydrate iron chloride(III), the other used reactants and the experimental conditions were the same. The previous synthesis was performed with the reactant Alfa Aesar-2 (AA2). The characteristics of RSNs obtained using different types of iron precursor are given in Table 2.

SEM images show that all experiments gave nanoclusters with an aggregated and spherical morphology, but RSNs have different mean sizes of nanoclusters, nanograins, and crystallites. The nanograin sizes determined from TEM images are always larger than the crystallite sizes. A correlation with the used iron chloride precursor is evident. Several hypotheses may explain such a discrepancy; our first hypothesis is the hydrolysis of iron chloride, as the latter is well known to be very sensitive to hydrolysis [24,25]. The iron precursors would display different hydration rates as a function of the aging time. Several groups also reported on the effect of water in polyol synthesis [26,27] and more precisely, in the nanocluster formation. Cao et al. [28,29] controlled the particle size and size distribution by adjusting the amount of water. They pointed out that the coordination of water molecules with iron ions is stronger than that of EG with iron. They observed a modification of the morphology with water: when water is added, the size of grains increases, and that of nanoclusters decreases. However, a direct link cannot be confirmed because these authors also increased the concentration of iron in their initial solution, and several groups have already confirmed a link between iron concentration and morphology [28].

Another problem encountered in these experiments was the random formation of iron carbonates within the final product (see details in SI: iron carbonate formation). The iron carbonate particles (siderite) with their characteristic brush morphology [30] were observed in the SEM images (Appendix A). As they are not identified by X-ray diffraction (except for one sample), these iron carbonates should represent less than 5 weight % of the sample (the necessary percentage to be detected by XRD). Iron carbonates are certainly formed out of CO_2_ produced by the thermal decomposition of urea into CO_2_ and ammonia [9]. Once formed, this FeCO_3_ cannot be decomposed further because of its stability over a wide range of temperatures. Similar carbonate formation was reported by Li et al. [31], who observed the intermediate formation of cobalt carbonates, CoCO_3_, which decompose to form cobalt ferrite at very high temperatures (>800 °C). It is worth noting that if a new (unaged) iron chloride precursor was used in the synthesis, no carbonates were detected. Therefore, the potential hydration of the reactant could be linked to the carbonates’ formation.

Thus, the above results could be explained by an aging effect resulting in the gradual hydration of different iron precursors. All iron chlorides were characterized to check if the iron chlorides were hydrolyzed or polluted by other substances. XRD, TGA, element analyses, and iron quantification experiments were performed and are detailed in the SI.

First, the color of the four iron precursors is different (Appendix A). Sigma 1 has a bright yellow color; Alfa Aesar-1 is darker and presents a brown-orange color rather than yellow; Alfa Aesar 2 is visually more similar to Sigma, and Acros Organics (AO1) is opaquer. The morphology of powders is also different: Sigma 1 and Alfa Aesar 2 (AA2) are less compact than Alfa Aesar 1 (AA1) and Acros Organics. The iron amount has also been determined by relaxometry measurements (Table 3). All commercial batches of iron chloride(III) hexahydrate contain less iron than expected. However, there is no strong correlation between the amount of iron and the mean RSN diameter (Table 3). The lower iron content is suggested to be due to a higher water content or hydrolysis rate of the iron precursors.

SEM images of the four reactants (Appendix A) showed the same morphology: a rough but uniform surface without grains or sheets. The associated EDS elementary analysis (Appendix A) showed, in addition to iron, chlorine, and oxygen, the presence of traces of some elements (aluminum and silicium), which do not seem to impact the RSN synthesis (see details in SI). Fe/Cl and Fe/O ratios have been extracted from these analyses (Table 3, Appendix A, Fe and Cl from FeCl_3_, and Fe and O from H_2_O). In theory, the atomic ratios for Fe/Cl of 0.33 and Fe/O of 0.165 are expected. The mean diameter of RSNs decreases with the increase in both ratios, showing the impact of these elements in the synthesis. The analysis of these results, detailed in SI, led us to conclude that a hydrolysis of iron chloride [24] has certainly occurred. However, as the iron precursors are in the form of big particles (SEM images in Appendix A), one might expect that the particles consist of a core of FeCl_3_·6 H_2_O with a hydrolyzed layer at the surface.

The hydrolysis of the precursors may be calculated (Appendix A), and the hydrolyzed species given in Appendix A are similar to those observed during the synthesis of the iron stearate precursor from iron chlorides: Fe(OH)_2.6_(H_2_O)_3.4_ [25]. Some differences are observed, suggesting a higher hydrolysis of Sigma and AA1 by comparison with other iron chlorides. However, these hydrolysis reactions do not allow explaining the observed ratios and are obtained by putting iron chlorides in direct contact with water, which is not the case here. A partial hydration may explain the decrease in the Fe/Cl ratio but not that of the Fe/O ratio, which in all cases remains in the range of 0.30–0.38. In fact, the O/Fe ratio is very close to 3 without a large amount of water and it suggests the formation of either Fe(OH)_x_(H_2_O)_y_ with a lower amount of water than a hydrolysis in water and even quite no water or Fe(OH)_x_Cl_y_ type compounds. The formation of iron oxychlorides cannot be excluded, nor can the beginning of the olation reaction lead to oxo-hydroxides such as FeOOH. The main problem is that there are numerous papers on the hydrolysis of iron chlorides in water but not on the aging of iron chlorides in powder form [24,25]. The AA2 and AO1 iron chloride samples are likely more “hydrolyzed”, which could explain the lower diameter of RSNs synthesized from these.

The different iron precursors have been characterized by XRD, FTIR spectroscopy, and TGA (Appendix A). XRD patterns (Appendix AA) show only the XRD peak characteristic of FeCl_3_·6H_2_O (PDR 00-033-0645), but FTIR spectra (Appendix AB) confirm the contamination of the precursors with silicon and the surface hydrolysis of the iron chlorides with a different hydrolysis rate. TGA experiments (Appendix AC) were performed to try to quantify the amount of water, but, if the water quantification remains difficult, TGA curves were shown to exhibit different characteristics (see SI on TGA experiments).

#### 3.1.3. Discussion on the Synthesis of RSNs

To conclude on the effect of the commercial origin of iron precursors: different commercial batches of FeCl_3_·6H_2_O led to RSNs with different mean diameters and nanograin sizes. The different characterizations performed showed that they are all hydrated and that a hydrolysis of some precursors cannot be excluded. This hydration/hydrolysis is certainly responsible for the different observed results. In an agglomerate of reactant particles, a gradual modification of the composition should be observed: from the initial iron chloride (in the center) to a more hydrated form at the surface. In agreement with the reported results on the effect of water on the RSN diameter [26], the hydration of the iron chloride precursor leads to a smaller diameter of RSNs. The variation of the nanograin size is not easy to explain, but it should also depend on the hydration rate. 

In the RSN formation mechanism reported in [11] and Appendix A, the coprecipitation of iron oxide nanoparticles occurs first due to ammonia resulting from urea decomposition, and then there is a heterogeneous nucleation and growth induced by the decomposition of an intermediate iron precursor resulting from the reaction of remaining iron chloride with EG. We may hypothesize that the different hydration rates of commercial precursors would affect these different steps. The hydration may favor the coprecipitation step leading to a higher amount of RSNs resulting from coprecipitation and there is thus not enough intermediate precursor to induce a heterogeneous nucleation. The intermediate precursor contributes only to the growth of the first-formed RSNs, which would explain their smaller diameter and also the different observed nanograin sizes. Further experiments would be needed to confirm these hypotheses.

Besides these observed differences in RSN diameter and nanograin sizes, we have evidenced another drawback of the iron precursors’ hydration, which is the formation of iron carbonates. This hydration affects their dissolution in ethylene glycol. Indeed, the effect of mixing time allowing to dissolve the reactants has been studied: with an “old” iron chloride flask, a mixing of the reaction mixture for overnight allowed avoiding the formation of carbonates. With “new” and “well stored” iron chlorides, overnight mixing was not necessary. This confirms another impact of the hydration rate of iron precursors.

Therefore, to obtain reproducible results, it is mandatory to store the flask in optimal conditions: under a vacuum atmosphere to avoid humidity and at a controlled temperature (maximum suggested temperature: 35 °C). RSN syntheses were then realized with a “new” FeCl_3_·6H_2_O precursor, stored under the previous conditions, and renewed every six months.

### 3.2. Optimization of the RSN Synthesis

A typical RSN synthesis (3 h of mix and 10.5 h of reaction) with the new iron precursor leads to the expected raspberry-shaped morphology (Appendix A) with oriented aggregates of nanograins (Appendix A). These nanoclusters, whose characterizations are detailed in SI, have a diameter of around 296 ± 35 nm with nanograins of ca. 25 nm, and a magnetite composition. Magnetic measurements confirmed their superparamagnetic behavior (no hysteresis) and led to a saturation magnetization value of ca. 90 emu·g^−1^, a value close to that of bulk magnetite (92 emu·g^−1^) [32]. These results are compared with those of Gerber et al. [23] and with previous results with the aged precursor in Table 4. The diameter and nanograin size are higher and would confirm the impact of the hydration of the precursor. The higher nanograin size leads to a composition as close to that of magnetite as expected and thus to a higher saturation magnetization [8,33].

As the objective is to synthesize RSNs with the highest specific surface area in order to capture as much phosphate as possible, different synthesis parameters have been tuned: the mixing times to dissolve reactants (3 h and overnight) and the reaction times (6 and 10.5 h).

**Effect of mixing time and reaction time.** SEM images confirmed the formation of aggregates of nanoparticles, and the characterization results are given in Table 5. The values of the lattice parameter confirm a magnetite composition for all samples. For experiments B, C, and D, nanoclusters of around 300 nm are obtained when, in synthesis A, RSN diameters are around 266 nm. In experiments A and B, with 3 hours of mixing and different reaction times, there is a strong correlation between the reaction time and the RSN size and grain size. This result agrees with the observations of Gerber et al., that longer reaction time produced the bigger RSNs with higher grain size [11]. Therefore, it explains the increase in the crystallite size with the reaction time, whatever the mixing time.

However, considering experiments C and D with different reaction times and conducted with a similar but longer mixing time (overnight), one may notice that RSNs have almost the same size and only the crystallite size has increased. This suggests that the overnight mixing led to RSNs, which reached their maximal diameter after 6 h of reaction while 10.5 h of reaction time is needed when the reactants’ mixing time is only 3 h.

The effect of the dissolution time (mixing time) of reactants is also visible by comparing experiments A and C, and B and D. After 6 h hours of reaction, the mixing time of 3 h (A) leads to smaller RSNs compared to that of 24 h (C). By contrast, 3 h of mixing and 10.5 h of reaction lead to similar RSN sizes as 24 h of dissolution and 6 h of reaction, but the grain size is higher. Therefore, the longer the mixing time the faster the reaction. These experiments show that the dissolution of the reactants is a very important step and affects the reaction kinetics and thus the characteristics of RSNs. 

Finally, Brunauer–Emmett–Teller (BET) measurements showed the specific surface area in the range of 17 and 30 m^2^·g^−1^. As expected, the RSNs with the smaller nanograin sizes display the highest values.

Therefore, the conditions leading to RSNs with the highest surface specific area are: 3 hours of mixing and 10.5 hours when the temperature plateau is reached (200 °C).

### 3.3. Synthesis of Doped Iron Oxide Nanostructures

The doping conditions have been optimized by taking care to preserve the raspberry morphology, to be able to remove the particles with a magnet and to obtain a high specific surface area.

#### 3.3.1. Doping of RSNs with Zinc (Zn-RSNs)

Zinc doping is interesting for modulating saturation magnetization. In the doped ferrite, zinc mostly occupies the tetrahedral sites in the spinel structure, and M_S_ would increase if x in Zn_x_Fe_2−x_O_4_ increases [34]. Nevertheless, after a given x, the measured magnetization decreases. Indeed, zinc ferrite (ZnFe_2_O_4_) has a spinel structure, and the absence of Fe^3+^ in Td sites (occupied by Zn) results in a weak antiferromagnetic material within Fe^3+^ and Fe^2+^ at Oh sites [35].

Preliminary experiments suggested that the zinc diffusion inside iron oxide in the RSNs would be longer, and thus the reaction time has been increased from 10.5 to 12 h and different iron/zinc ratios have been tested. In all cases, the SEM images (Figure 3(A.1–A.3)) confirm the formation of clusters of NPs with a partial conservation of the raspberry morphology. All ratios produced nanoclusters of similar size (300 ± 103 nm, 280 ± 86 nm, and 272 ± 46 nm, respectively). However, the 1.2:1 and 2:1 syntheses led to a larger particle size distribution. EDS analysis further showed a very low (<1%) amount of zinc for these syntheses. 

In order to enhance the zinc doping, the initial ratio of Fe-Zn was increased to 9:1. SEM images confirmed the RSN morphology, and EDS analysis showed an increase in the zinc amount (3%) even if it is always lower than expected. XRD patterns showed only the characteristic XRD peaks of a spinel phase but no modification of the lattice parameter (respectively: 8.391; 8.391 and 8.388 Å) confirming the low Zn doping. The calculated crystallite size was 32.6 nm for the last sample, this value corresponds to the grain size of the previous magnetite RSNs. The low amount of zinc in the doped RSNs confirmed the reported difficulty in doping magnetite RSNs with zinc [19].

From these experiments and reported results, we conclude that the difficulty is the complete dissolution of the zinc chloride precursor. In future experiments, other zinc precursors will be tested, as well as varying the mixing time.

#### 3.3.2. Doping of RSNs with Cobalt (Co-RSNs)

The ferrite CoFe_2_O_4_ drew attention because, in its bulk state, it is a well-known hard magnetic material with a high coercivity (H_C_), a high Curie temperature (520 °C), a moderate saturation magnetization (80 emu·g^−1^) and a high anisotropy constant (2.65 × 10^6^–5.1 × 10^6^ erg·cm^−3^). Stoichiometric CoFe_2_O_4_ NPs would present a higher saturation magnetization due to the presence of Co^2+^ ions in the octahedral sites of the spinel structure: [(Fe^3+^)_Td_(Co^2+^Fe^3+^)_Oh_](O^2−^)_4_ [36]. However, at the nanosize, Co^2+^ atoms are observed to replace also some Fe^3+^ atoms in tetrahedral sites leading to an inverse spinel structure: (Co_x_Fe_1−x_)_Td_[Co_1−x_Fe_1+x_]_Oh_O_4_, where x depends on the synthesis conditions [37].

Following the previous protocol (3 h of mixing), the RSN synthesis has been performed with two different Fe:Co ratios. For a Fe:Co ratio of 1:1, RSNs with a mean size of 292 ± 76 nm (Figure 4(A1)) are observed, but the size distribution is quite broad. With the Fe:Co ratio 2:1 (Figure 4(A2)), the RSN size distribution is narrower and the diameter smaller: 224 ± 26 nm. Some of the RSNs are hollow, and at the top of the SEM image, metal carbonate particles are identified. This carbonate formation is random and linked to the reactant dissolution step. Even if the SEM-EDS analysis showed a higher Co doping amount than with zinc (from 7 to 12 atomic %), the presence of carbonates affects this value. To avoid carbonate formation, the mixing time and the reaction time of 21 and 30 h have been increased to favor the reactants’ dissolution in the reaction media. SEM images (Figure 4B) showed RSNs with a close mean diameter: 287 ± 25 nm for 10.5 h, 262 ± 25 nm for 21 h, and 267 ± 22 nm for 30 h). However, for the synthesis for 10.5 h, nanograins appear smaller and the surface is less rough than the others. EDS analysis shows a decrease in the cobalt amount (7–7.5 %) (no carbonate was detected). From the three experiments, the reaction time appears to have no effect on the cobalt doping amount.

Figure 4C displays the XRD pattern of the last sample (stirred overnight and 10.5 h of reaction time); those of other samples are very similar and not presented here. The XRD peaks are related to a spinel structure (maghemite, magnetite) without the presence of another phase (same observation for the sample containing carbonates). The lattice parameter was calculated to be 8.41 Å, a slightly higher value than that of bulk magnetite; this increase can be related to the insertion of cobalt in the structure. The crystallite sizes are: 17, 27.8, and 36 nm for Co-RSNs with reaction times of 10.5, 21, and 30 h, respectively. The IR spectrum confirms the presence of the magnetite phase (inner image in Figure 4D) and the presence of EG traces. SQUID measurement at 300 K of the sample B.1 (with the smallest nanograin size) led to a saturation magnetization of 83.5 emu·g^−1^ for Co_x_Fe_3−x_O_4_ vs. 83.8 emu·g^−1^ for similar magnetite RSNs. As for other cobalt-doped iron oxide NPs, a hysteresis curve was observed, confirming the cobalt doping with a coercivity value of 400 Oe (Figure 4E). 

#### 3.3.3. Doping of RSNs with Aluminum (Al-RSNs)

Aluminum has been shown to be a suitable doping element to enhance phosphate removal. However, aluminum is a non-magnetic element, and Al substitution decreases Ms because, unlike Zn, Al^3+^ substitutes for Fe^3+,^ not Fe^2+^. So, to keep the possibility of “magnetic decantation”, the magnetite RSNs should be doped with a small amount of aluminum. The doping of magnetite nanostructures with aluminum should be easier than with Zn and Co because the ionic radius of Al and Fe^3+^ is quite close and their valence is the same (+3). For the doping process, a reaction time of 10.5 hours and a mixing time of 3 hours were tested, as well as three Fe:Al ratios. SEM images (Figure 5) show nanostructures with the morphology of undoped RSNs, but a broad size distribution is noticed with a mean diameter of 269 ± 45 nm for the last sample. EDS analyses confirm the success of the aluminum doping, with quite high doping amounts increasing with the amount of introduced Al. 

However, during the washing step, samples with an iron quantity in the Fe:Al ratio of 2 to 7 were difficult to decant magnetically. To facilitate this step, samples were centrifuged. However, it confirms that aluminum doping modifies the magnetic properties (in particular, saturation magnetization). The only sample that could be magnetically decanted was the last one (Fe:Al 10:1). Thus, this sample, called Al-RSNs, was characterized and then used for phosphate removal experiments.

XRD patterns (Figure 5B) confirm the preservation of the iron oxide spinel structure without the presence of other phases. The XRD peaks are broader for RSNs, with high Al content suggesting a loss of crystallinity or smaller crystallite sizes. The lattice parameter, 8.389 Å, is lower than that observed usually with undoped RSNs but would confirm the aluminum insertion in the spinel structure. The crystallite size is calculated to be 19.5 nm (Figure 5(A5)), suggesting that the Al doping leads to a smaller crystallite size. The highest surface specific area measured for this sample (40 m^2^/g) confirmed the smallest nanograin size of Al-RSNs compared to undoped RSNs. The magnetization curve at 300 K is characteristic of superparamagnetic behavior, and the saturation magnetization (Figure 5D), 64 emu·g^−1^ for Al_x_Fe_3−x_O_4_, is lower than that of undoped RSNs, confirming the doping of magnetite by Al and previously reported results of other groups [15,38,39]. 

### 3.4. Phosphate Removal Using Undoped and Al-Doped Iron Oxide Nanostructures 

RSNs, used for these experiments, have a mean diameter of 319 nm and a mean nanograin size of 29 nm (surface specific area = 27 m^2^ g^−1^). RSNs doped with 10% of Al have a mean diameter of 269 nm, a mean nanograin size of 19.5 nm, and a surface specific area of 40 m^2^·g^−1^.

To ensure a high adsorption of phosphate at the surface of iron oxide nanostructures, it is important to favor electrostatic interactions between phosphate and the iron oxide surface. The isoelectric point (IEP) of RSNs is about 5.6, as depicted in Appendix A, and is slightly shifted by comparison with the reported IEP of iron oxide, usually in the range 6–7 [40]. This is certainly due to the washing process, which does not completely remove all reactants (especially ethylene glycol) from the RSN surface. For phosphates, an increase in pH leads to a change in the main species in the solution from H_3_PO_4_ to PO_4_^3–^ (Appendix A). Thus, the suitable electrostatic interactions between deprotonated phosphates and iron oxide surfaces are for pH < 5.5. Thus, optimal adsorption conditions would be around pH 3, as the phosphates are deprotonated (negatively charged) and the iron oxide surface is positively charged. However, in water depollution conditions, the pH is often around 7. At this pH, the solution contains phosphates in H_2_PO_4_^−^ and HPO_4_^2−^ form when the surface of RSNs is slightly positively charged. At such a pH, the adsorption maximum is expected to be much lower than at pH 3.

The zeta potential curve of Al-RSNs as a function of pH in Appendix A shows clearly that the insertion of aluminum into the spinel iron oxide structure induces a shift of the ZP curve towards higher pH. The value of the isoelectric point is shifted from 5.6 to 7.2, and therefore, at pH 7, the electrostatic interactions would be stronger, and the phosphate capture should be favored.

#### 3.4.1. Phosphate Removal from Undoped and Al-Doped RSN Solutions as a Function of Time 

In the first experiment, RSNs were put in contact with phosphate solutions at various times. Figure 6 shows that the maximum adsorption is reached after one hour for RSNs and 3 h for Al-RSNs. Then, the curve tends to reach a saturation plateau. Some adsorption results with Fe- and Al-Fe-based materials are given in Appendix A. In other studies of Al-Fe-based materials (Appendix A), the maximum adsorption is reached after 5 h [15,38,40,41,42]. So, at first sight, the Al-RSNs synthesized in the present work would ensure a faster phosphate caption.

For most other iron oxide-based nanomaterials (Appendix A), the maximum phosphate caption is reached after 5 to 15 h. The only nanomaterial that shows a caption time similar to ours is the iron hydroxide eggshell of Mezenner et al. [43]. These authors observed a maximum adsorption after 3 hours. From these results, we may conclude that the iron oxide RSNs without Al adsorb the phosphates at the fastest rate. 

The adsorption curves have been fitted with two kinetic equations. In Figure 6, we can observe that, for RSNs, both models seem to fit quite well the experimental data and give similar kinetics parameter values (Table 6). The R² value of the pseudo-second-order model is the highest and closest to 1. This model suggests that chemical sorption is the rate-determining parameter [44,45,46]. More precisely, the pseudo-second-order model describes the adsorption in two steps. The first step is a rapid adsorption on the surface of the adsorbent with abundant vacant adsorption sites. The second step is a slower diffusion to finish the saturation of adsorbent sites [47]. This would be in agreement with a quick adsorption of phosphate on iron sites and then the diffusion of phosphates to fill the remaining sites. It is difficult to compare our results with published results because the adsorption is strongly dependent on the initial concentration, the amount of adsorbent introduced, and the temperature. To remove phosphates from water, different iron oxide-based materials have been designed (Appendix A): for example, iron oxide-impregnated strong base anion exchange resin [48], a hybrid fibrous exchanger containing hydrated ferric oxide nanoparticles [49], or a hybrid anion exchanger containing triethylamine functional groups and hydrated Fe(III) oxide nanoparticles [50]. Our results are compared with studies performed on iron oxide (nano)materials (Appendix A). In Appendix A, most of the data also fit better with the pseudo-second-order model. The comparison of our RSNs to the other iron-based materials shows that the *k_2_* value is the highest (5.28 h^−1^), which confirms that our RSNs can adsorb phosphates faster. 

For Al-RSNs, both models seem to fit the data quite well, but the pseudo-first-order model fits the experimental data better. Such a model suggests a diffusion-controlled process. Few works have been published on phosphate capture by iron oxide doped with aluminum nanoparticles, and the main results are summarized in Appendix A. The results in Appendix A show that most kinetics results are fitted with the pseudo-second-order model. 

To conclude, experimental results showed a maximum adsorption of phosphate after 2 hours with RSNs and 3 hours with Al-RSNs, which can be considered a “fast adsorption”. 

#### 3.4.2. Phosphate Removal from RSN and Al-RSN Solutions as a Function of the Initial Phosphate Concentration 

The adsorption curves as a function of pH and phosphate concentrations are given in Figure 7. As expected, and in agreement with already reported results, the adsorption is more efficient at pH 3 than at pH 7 due to more favorable electrostatic interactions. This trend was also reported by several groups with different materials: magnetite [7], goethite [51], aluminum oxide hydroxide [52], or MnO_2_ [53].

If we compare the phosphatation behavior of RSNs and Al-RSNs (Figure 7A,B), the capture of phosphates with Al-RSNs appears slower, and the plateau is reached for an initial phosphate concentration higher than 60–70 P-mg·L^−1^ whereas for RSNs, it was reached at 50 P-mg·L^−1^. For pH 3, the maximum adsorption amount of Al-RSNs is 15.5 P-mg·g^−1^, and at pH 7, the value is about 10 P-mg·g^−1^. These values are higher than those with RSNs, in agreement with the higher surface specific area of Al-RSNs.

Concerning these isotherms, curves are of the «L» type. Such an “L” isotherm confirms the strong affinity of phosphate for iron oxide surfaces and a progressive saturation of the surface when the concentration increases. These curves, obtained in water at pH 3 and 7, were fitted with three different equilibrium models. The fitting curves and the adsorption parameters are presented in Appendix A and one may observe that these models do not allow for fitting well the experimental curves. Nevertheless, one may notice that the Langmuir model seems to be the most suitable model. This model suggests that the adsorption sites do not interact between them and that their energies are equivalent. More experimental data are needed in the curves to conclude the most suitable model and also to consider parameters resulting from the fitting. 

To compare our results with the reported adsorption studies listed in Appendix A, we have to be careful again, as this adsorption strongly depends on the experimental conditions: media, temperature, initial amount of adsorbent, and concentration of phosphates. The conditions and adsorption results are summarized in Appendix A for RSNs and Appendix A for Al-RSNs. In Appendix A for RSNs, the models that fit better are the Langmuir and Freundlich ones. It confirms the fitting of the previous RSN isotherm curve in water with the Langmuir model. The smaller adsorbed amount of phosphates with RSNs than with the other iron materials may be explained by their lower surface specific area. In Appendix A for Al-RSNs, the model that better fits the curve is the Langmuir one. It confirms that the Al-RSN isotherm curve in water could be fitted with this model. This model suggests that the adsorption sites do not interact among themselves and that their energies are equivalent.

#### 3.4.3. Adsorption Amount

The maximum adsorption of RSNs at pH 3 was calculated to be 8.8 P-mg·g^−1^. At physiological pH in water, the maximum adsorbed value drops to 4.1 P-mg·g^−1^. At pH 7, the maximum adsorption value for Al-RSNs is 10 P-mg·g^−1^, at pH 3, it rises to 15.5 P-mg·g^−1^. Both values are significantly larger than for RSNs (4.1 P-mg·g^−1^ and 8.8 P-mg·g^−1^), but the surface specific area of Al-RSNs is also higher. Thus, one reason for Al doping is to increase the surface specific area, allowing a higher phosphatation capture per g of material. One may notice that the maximum adsorption values obtained with Al-RSNs are quite high compared to those reported in the literature (Appendix A) except for the studies of Xu et al. [15].

Our adsorption amounts were compared to those of Daou et al. [7], who have used nanoparticles with a close surface specific area (30 m^2^·g^−1^) and similar experimental conditions (room T, t = 24 h, adsorbent: 1 g·L^−1^, pH 3 and pH 7). The adsorption amounts with RSNs and Al-RSNs are higher than those reported by Daou et al.: at pH 3, they measured 5.2 P-mg·g^−1^ (3.26 P-molecule·nm^−2^) and at pH 7, 1.5 P-mg·g^−1^ (1.02 P-molecule·nm^−2^). Therefore, RSNs and even Al-RSNs allow a higher adsorption of phosphates on their surfaces than iron oxide NPs. It could be explained by a different phosphate complex (i.e., monodentate, bidentate), the formation of a second phosphate layer, or a more favorable nanostructuration of RSN (higher curvature of grains or stronger adsorption at interfaces).

## 4. Conclusions

This study investigated the use of iron oxide nanostructures as an adsorbent to improve phosphate removal for water depollution. The synthesis of iron oxide and aluminum-doped ferrite raspberry-shaped nanostructures (RSNs and Al-RSNs) was optimized using a solvothermal–polyol method. The impact of synthesis parameters such as the nature of the iron precursor, the reaction time, and the mixing/solubilization step has been demonstrated. Then, the phosphate removal properties of these nanostructures were tested by studying their adsorption capacity and kinetics. The RSNs showed great affinity for phosphate, with a maximum adsorption capacity of 4.1 P-mg/g at pH 7 and 8.8 P-mg/g at pH 3. The Al-RSNs allowed a higher enhancement of adsorption capacity, with 10 P-mg/g at pH 7 and 15.5 P-mg/g at pH 3. The Al doping of RSNs shifted the IEP of RSNs and allows thus favorable electrostatic interactions. In addition, the surface specific area of Al-RSNs was higher. In addition, it was demonstrated that the phosphate maximum absorption was reached in less than 3 h for both undoped and Al-doped RSNs. These overall results showed that such nanostructures are promising for phosphate removal in water depollution.

## Figures and Tables

**Figure 1 nanomaterials-13-00587-f001:**
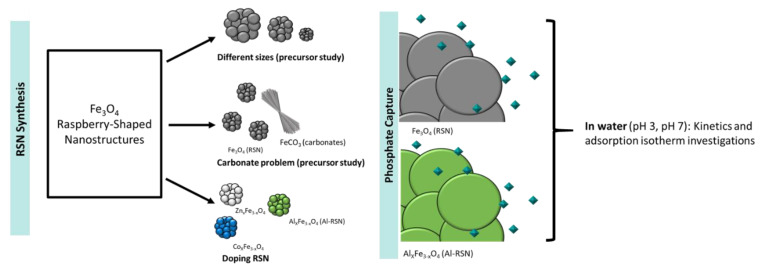
General concept schematizing the different performed investigations on the synthesis of the raspberry-shaped nanostructures and on the evaluation of the RSN and the Al-RSN materials for phosphate removal.

**Figure 2 nanomaterials-13-00587-f002:**
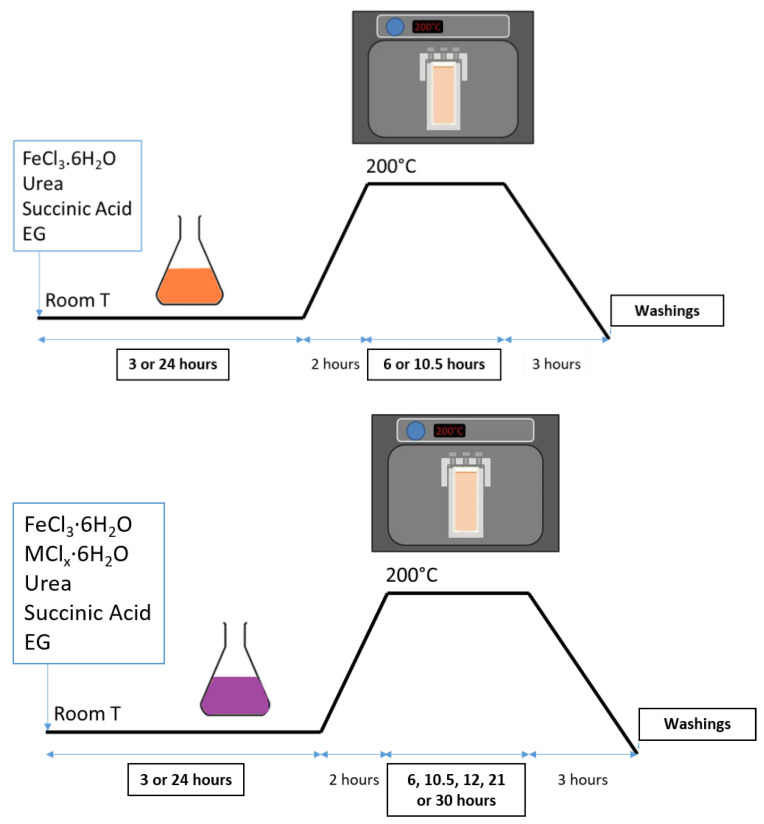
Synthesis conditions of RSNs (top) and of doped RSNs (down). EG: ethylene glycol.

**Figure 3 nanomaterials-13-00587-f003:**
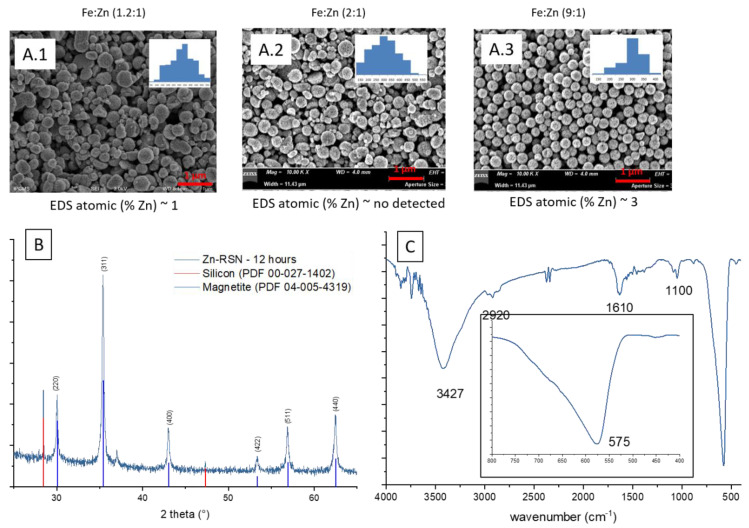
**(A.1**–**A**.**3**) SEM images of all zinc-doped RSNs (inner image: size distribution); (**B**) XRD pattern; (**C**) IR spectrum of zinc-doped RSNs (9:1) (mixing time: 3 h; reaction time 12 h) (inner image: zoom on the Fe–O band characteristic of slightly oxidized magnetite).

**Figure 4 nanomaterials-13-00587-f004:**
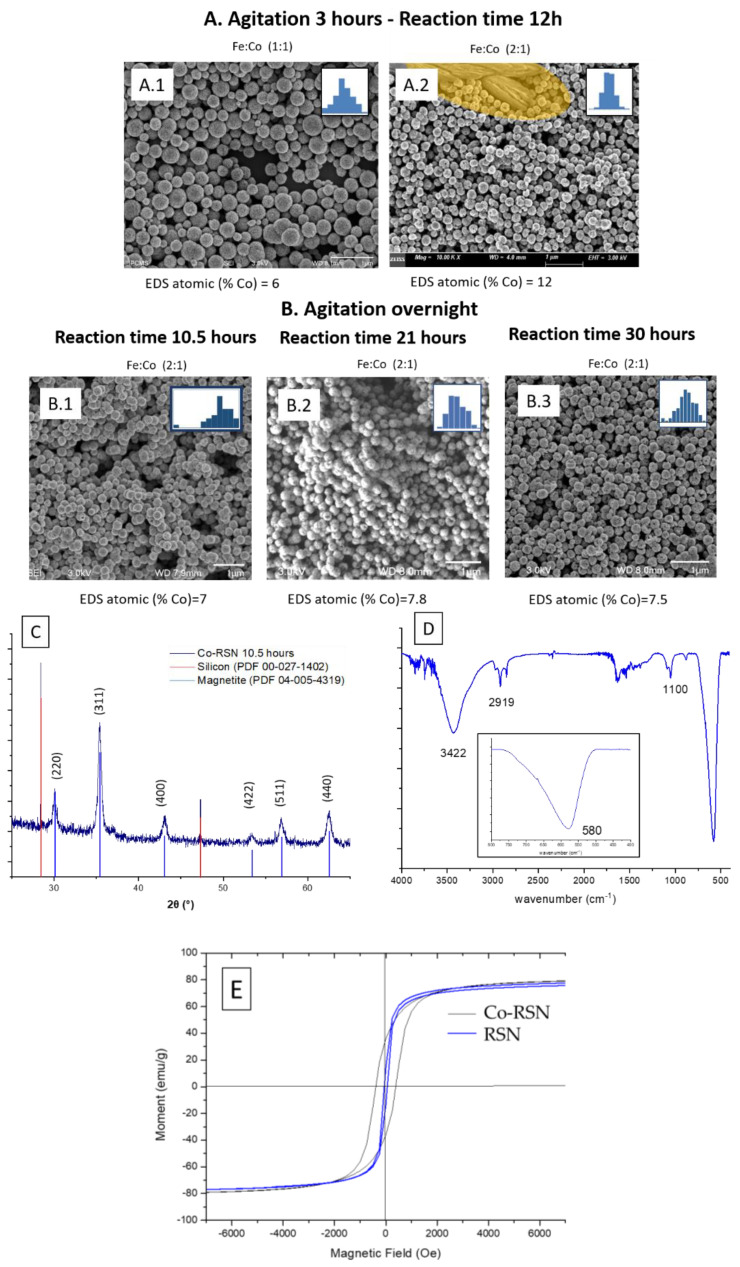
SEM images of cobalt-doped RSNs. (**A**) Experiments with 3 h mixing (golden shading in A2 shows the carbonate particles and inner images = SEM size distribution). (**B**) Experiments with overnight mixing. (**C**) XRD pattern. (**D**) FTIR spectrum (inner image = zoom on Fe–O bands showing one band at 580 cm^−1^ characteristic of the magnetite phase). (**E**) Magnetization curves of the undoped and cobalt-doped (B.1) RSN at 300 K.

**Figure 5 nanomaterials-13-00587-f005:**
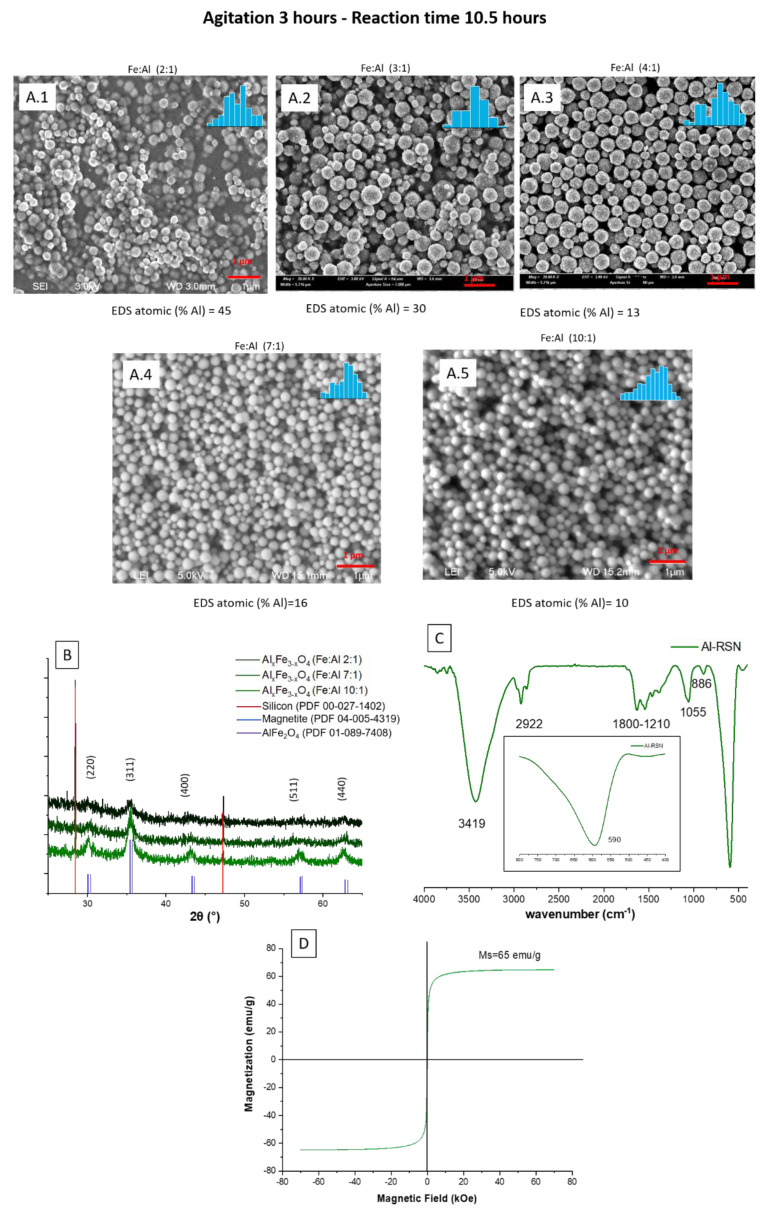
(**A.1**–**A.5**) SEM images of Al-RSNs (inner image: size distribution); (**B**) XRD pattern as a function of the Fe:Al ratio; (**C**) FTIR spectra (inner image: zoom of Fe–O band characteristic of a slightly oxidized magnetite phase); (**D**) magnetization curve of Al-RSNs at 300 K.

**Figure 6 nanomaterials-13-00587-f006:**
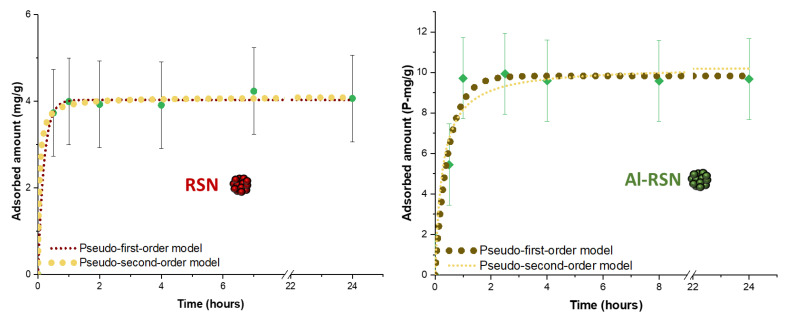
Adsorbed amount of phosphate by RSNs (left) and Al-RSNs (right) in a phosphate solution (50 P-mg·L^−1^) at pH 7 for different durations.

**Figure 7 nanomaterials-13-00587-f007:**
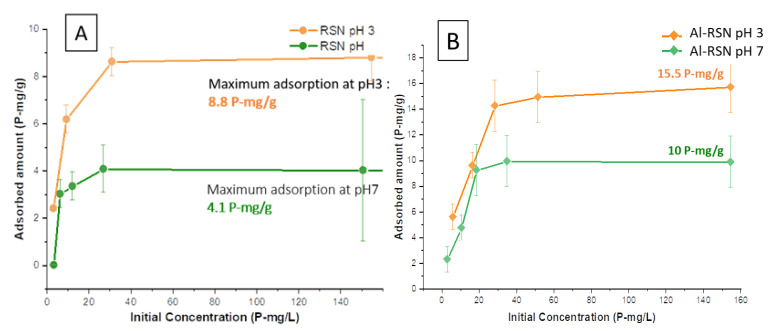
Adsorbed amount of phosphate in different media for 24 h by (**A**) RSNs and (**B**) Al-RSNs. In orange, in water at pH 3; in green, in water at pH 7.

**Table 2 nanomaterials-13-00587-t002:** Mean crystallite size (deduced from XRD patterns) and mean diameter and nanograin size of RSNs as a function of the used iron precursor and representative SEM images of the synthesis.

Brand and Lot	Sigma-1	Alfa Aesar-1	Alfa-Aesar-2	Acros Organics-1
**RSN size (nm)**	291 ± 52	296 ± 35	157 ± 42	~100
**TEM nanograin size (nm)**	38 ± 7	26 ± 6	30 ± 6	27 ± 9
**XRD crystallite size (nm)**	32.4	19.9	20.2	25.4
**TEM image** **(scale bar = 1 μm)**	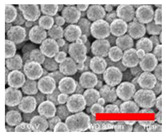	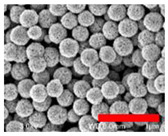	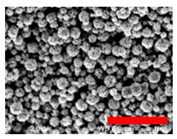	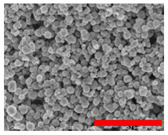

**Table 3 nanomaterials-13-00587-t003:** Fe/Cl and Fe/O atomic ratios determined by SEM-EDS and amount of iron in the different iron precursors determined by relaxometry measurements.

Iron Precursor	Sigma 1	Alfa Aesar 1	Alfa Aesar 2	Acros Organics 1
**RSN diameter (nm)**	291 ± 52	296 ± 35	157 ± 42	100 ± 50
*Iron amount determined by relaxometry measurements*				
**Expected iron weight (mg)**	1	1	1	1
**Measured iron weight (mg)**	0.89	0.93	0.94	0.93
*SEM-EDS analysis: Theoretical Fe/Cl = 0.33 and Fe/O = 0.17*				
**Fe/Cl**	0.35	0.36	0.38	0.44
**Fe/O** FeCl_x_·yH_2_O	0.33 FeCl_2.86_·3H_2_O	0.30 FeCl_2.78_·3.3H_2_O	0.38 FeCl_2.63_·2.6H_2_O	0.37 FeCl_2.27_·2.7H_2_O

**Table 4 nanomaterials-13-00587-t004:** Comparison of the RSNs synthesized with different iron precursors.

	Gerber et al. (Sample RSN25)	First RSNs (AA2 Precursor)	New RSNs (New Precursor)
**Size of RSN (nm)**	245 ± 12	157 ± 42	296 ± 35
**TEM nanograin size (nm)**	25 ± 3	30 ± 6	38 ± 10
**XRD crystallite size (nm)**	15.5 ± 0.2	20.2 ± 0.3	30 ± 0.2
**Lattice parameter (** **Å)**	8.39 ± 0.01	8.39 ± 0.01	8.40 ± 0.01
**Saturation magnetization (emu g^−1^)**	78	70	90

**Table 5 nanomaterials-13-00587-t005:** RSN synthesis of experiments A, B, C, and D with 3 or 24 h of mixing and 6 or 10.5 h of reaction time at the highest reaction temperature (standard deviation for lattice parameter: ±0.01).

Experimental Conditions	A 3h Mixing, 6h Plateau	B 3 h Mixing, 10.5 h Plateau	C Overnight Mixing, 6 h Plateau	D Overnight Mixing, 10.5 h Plateau
**SEM image** **(scale bar: 100 nm)**	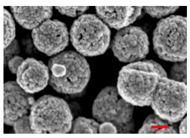	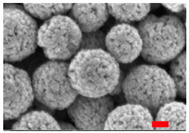	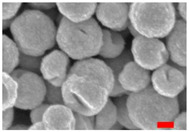	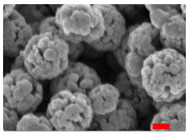
**RSN size (nm)**	~266	~317	~320	~310
**Nanograin size (nm)**	24 ± 5	29 ± 8	36 ± 12	43 ± 10
**Lattice parameter (** **Å)**	8.41	8.39	8.40	8.40
**Specific surface area (m^2^ g^−1^)**	24.2	27	18.9	17.8

**Table 6 nanomaterials-13-00587-t006:** Kinetic parameters for pseudo-first-order and pseudo-second-order models for RSNs and Al-RSNs.

	Pseudo-First-Order			Pseudo-Second-Order		
	*q_e_* (P-mg·g^−1^)	*K_1_* (h^−1^)	R²	*q_e_* (P-mg·g^−1^)	*K_2_* (g·P-mg^-1^·h^−1^)	R²
**RSN**	4.0 ± 0.1	5.17 ± 0.84	0.951	4.1 ± 0.1	5.28 ± 2.18	0.996
**Al-RSN**	9.8 ± 0.3	2.04 ± 0.35	0.974	10.3 ± 0.6	0.37 ± 0.18	0.942

## Data Availability

The data that support the findings of this study are available in the supplementary material of this article. The data are also available on request from the corresponding author. The data are not publicly available because the authors want to keep priority for conference presentations.

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
