# Peer review of "Phosphate Capture Enhancement Using Designed Iron Oxide-Based Nanostructures"

_nanomaterials, 2023, doi:10.3390/nano13030587_

Round 1
Reviewer 1 Report
The authors present an excellent, complete research work with great designed figures. Implementing small changes, in my opinion, it could be published in Nanomaterials.
Please, check the presence of double spaces.
Table 1 should compare a greater number of articles, thus providing a broader vision of the state of the art.
Line 169. Check chemical equations (Sn oxidation states are written as subscripts).
Lines 233-235. The format does not match the rest of the article, please modify.
Please, check phrase-citation separation (e.g., line 321).
Could you rename section III.1 c) in some other way? As the conclusions section is so widespread in the scientific literature, I think it introduces uncertainty to the reader.
Figure 6 is the one that gives me the most doubts. I think more experimental points should have been taken, especially at low times, to check a better fit to the kinetic curves.
Figure 7 should be retouched to improve its quality, since figures 7C and 7D cannot be appreciated, which are intuited and which, by the way, do not appear in the text.
Author Response
Manuscript ID: nanomaterials-2149308
Type of manuscript: Article
Title: Phosphate capture enhancement using designed iron oxide-based
nanostructures
Authors: Paula Duenas-Ramirez, Chaedong Lee, Rebecca Fedderwitz, Antonia
Regina Clavijo, Débora Pires Porto Barbosa, Maxime Julliot, Joana Vaz-Ramos,
Dominique Bégin, Stéphane Le Calvé, Ariane Zaloszyc, Philippe Choquet,
M.A.G. Soler, Damien Mertz, Peter Kofinas, Yuanzhe Piao, Sylvie Bégin-Colin *
The authors thank the reviewers for their very constructive remarks and comments, which have contributed to improve the quality of our publication. Please find below a detailed reply to the comments along with the text, which was added to the revised version.
Best regards
The authors
Reviewer Comments to Author:
Reviewer 1.
The authors present an excellent, complete research work with great designed figures. Implementing small changes, in my opinion, it could be published in Nanomaterials.
Please, check the presence of double spaces.
Response : the authors thank the reviewer for the constructive remark and the text is now in the Nanomaterials template.
Table 1 should compare a greater number of articles, thus providing a broader vision of the state of the art.
Response : The state of the art in Table 1 has been updated
Line 169. Check chemical equations (Sn oxidation states are written as subscripts).
Response : The chemical equation has been corrected.
Lines 233-235. The format does not match the rest of the article, please modify.
Response : The format has been updated.
Please, check phrase-citation separation (e.g., line 321).
Response : The phrase citation separation has been corrected.
Could you rename section III.1 c) in some other way? As the conclusions section is so widespread in the scientific literature, I think it introduces uncertainty to the reader.
Response : The section II.1.C has been renamed: “c) Discussion on the synthesis of RSNs”
Figure 6 is the one that gives me the most doubts. I think more experimental points should have been taken, especially at low times, to check a better fit to the kinetic curves.
Response: The quantification of phosphate is quite complex as explained in the experimental part and the standard deviation on measurements can be quite high especially for short adsorption times. Shorter times were also more difficult to control. That’s why we introduced only relevant measurements.
Figure 7 should be retouched to improve its quality, since figures 7C and 7D cannot be appreciated, which are intuited and which, by the way, do not appear in the text.
Response: The Figure 7 has been updated.

Reviewer 2 Report
The mansuscript reports on a very interesting and detailed study of raspberry-shaped nanoclusters (RSN) of iron oxides, included doped ones, as an agent for phosphorus removal from the liquid media. I would particularly commend authors' meticulous study of the effect of iron chrolide precursor hydration upon the properties of iron oxide RSNs.
My major concern with this paper is the presentation of the material. In my opinion, parts concerned the synthesis of Zn and Co doped iron oxide could well be relegated to the supplementary materials, or even omitted, since the phosphorus caption was investigated only for pure and Al doded iron oxide RSNs. On the other hand, the full characterization of the pure iron oxide RSN used fro phosphorus caption experiments should appear in the main body of the paper.
Some experimental methods (magnetics, BET) are not described in the respective section.
English style of the paper needs a substantial polishing. I inserted some suggestions and comments directly into manuscript files.

Author Response
Manuscript ID: nanomaterials-2149308
Type of manuscript: Article
Title: Phosphate capture enhancement using designed iron oxide-based
nanostructures
Authors: Paula Duenas-Ramirez, Chaedong Lee, Rebecca Fedderwitz, Antonia
Regina Clavijo, Débora Pires Porto Barbosa, Maxime Julliot, Joana Vaz-Ramos,
Dominique Bégin, Stéphane Le Calvé, Ariane Zaloszyc, Philippe Choquet,
M.A.G. Soler, Damien Mertz, Peter Kofinas, Yuanzhe Piao, Sylvie Bégin-Colin *
The authors thank the reviewers for their very constructive remarks and comments, which have contributed to improve the quality of our publication. Please find below a detailed reply to the comments along with the text, which was added to the revised version.
Best regards
The authors
Reviewer 2
The mansuscript reports on a very interesting and detailed study of raspberry-shaped nanoclusters (RSN) of iron oxides, included doped ones, as an agent for phosphorus removal from the liquid media. I would particularly commend authors' meticulous study of the effect of iron chrolide precursor hydration upon the properties of iron oxide RSNs.
My major concern with this paper is the presentation of the material. In my opinion, parts concerned the synthesis of Zn and Co doped iron oxide could well be relegated to the supplementary materials, or even omitted, since the phosphorus caption was investigated only for pure and Al doded iron oxide RSNs.
Response : The authors thank the reviewer for their constructive remarks.
Concerning the doping part, the doping of RSN is very difficult and few papers reported on this doping or do not really proved that the doping was successful. Therefor, we think that this doping part is quite important and shows. Therefore, we propose to maintain this doping part but if the reviewer wants that we remove it, we will do it.
On the other hand, the full characterization of the pure iron oxide RSN used fro phosphorus caption experiments should appear in the main body of the paper.
Response : the characterizations are quite standard and already widely reported in the text above. In addition, they will take a lot of place in the main text. The main important characteritics for this study are detailled in the text and in Table 4 where all RSNs are compared. These informations allowed understanding the different properties.
Some experimental methods (magnetics, BET) are not described in the respective section.
Response : these characterization techniques have been described in the materials and methods part.
English style of the paper needs a substantial polishing. I inserted some suggestions and comments directly into manuscript files.
Response : We thank the reviewer very sincerely for the very careful proofreading and the very constructive corrections that were made.
Comments in manuscript file :
Lines 243-244 : Remarks : « Higher oxidation would result in smaller lattice parameter, which is not observed ».
Response : in fact in RSNs, the lattice parameter is always higher to that of bulk magnetite and would be due to defects and strains induced by the oriented agreagation. Therefore, it is difficult to evaluate the composition of RSNs from consideration of the lattice parameter.
Line 259 : Remark : « It would be logical to place the data related to the first synthesis in the leftmost column. List “ageing” times when known ».
Response : the table include SEM images and it takes a lot of place. In addition, SEM and TEM images have already been published and both samples have been compared in the previsous paragraph.
Lines 306-307 : Unclear phrase.
Response : the sentence has been modified : The lower iron content is suggested to be due a higher water content or hydrolysis rate of the iron precursors.
Line 325 : ??? The formula is for iron hydroxide
Response : yes because we have a hydrolysis reaction here and not hydration.
Line 372 : Make what less soluble in EG? Please clarify.
Response : the sentence has been modified to be clearer : This hydration affects their dissolution in ethylene glycol.
Line 385 : I suggest that these characteristics would be better to be shown in the main body of the paper, since this material serves a reference for the comparison with doped iron oxides whose properties are show in the main text.
Response : the characterizations are quite standard and already reported in the text above. In addition, they will take a lot of place in the main text. The main important characteristics are detailled in the text and in Table 4 where all RSNs are compared. These informations allowed understanding the different properties.
Line 402 : What about saturation magnetization?
Response : we have not determined the saturation magnetizations for this particular study.
Line 403 : Show these with the respective uncertaintes. 8.41 Å is a bit too high for magnetite,
isn’t it?
Response : as we already have explained it, the defect and strain resulting from the oriented agregation in RSNs lead to a lattice parameter higher than the lattice parameter of the bulk magnetite phase.
Line 455 : These values are lower than even for stoichiometric magnetite! It can be concluded therefore that Zn doping was not successful.
Response : the lattice parameter of bulk magnetite is of 8.39
Comments in manuscript SI file :
We thank the reviewer for his/her carefull reading and corrections : The english style has been corrected
Page 4 : Why do you need this phrase : « These impurities show up a clue”
Response : the authors agree with the reviewer remark and this sentence has been removed.
Page 5 : What is it supposed to mean?: « However, a link between the synthesized RSN cannot be concluded from these data »
Response : We wanted to say that the impurities do not allow explaing the observed difference between the different commercial batches. The sentence has been modified for more clarity : However, all reactants contain similar impurities and thus that does not allow explaining the observed differencies in characteristics of RSN.
